# Efficacy and Safety of Fig (*Ficus carica* L.) Leaf Tea in Adults with Mild Atopic Dermatitis: A Double-Blind, Randomized, Placebo-Controlled Preliminary Trial

**DOI:** 10.3390/nu14214470

**Published:** 2022-10-25

**Authors:** Tatsuya Abe, Yukari Koyama, Kosaku Nishimura, Aya Okiura, Toru Takahashi

**Affiliations:** Toyo Institute of Food Technology, 23-2, 4-Chome, Minami-Hanayashiki, Kawanishi 666-0026, Hyogo, Japan

**Keywords:** atopic dermatitis, Eczema Area and Severity Index (EASI), RCT, functional food, *Ficus carica*

## Abstract

Atopic dermatitis (AD) is a chronic, recurrent pruritic skin disease with repeated remissions and exacerbations. Various factors, such as allergies, skin conditions and lifestyle, combine to cause AD, making it difficult to cure completely. Although AD symptoms are suppressed with medications, this is a long-term effort and burden on patients. Thus, safer drugs and alternatives are needed. We previously found that consumption of tea prepared from fig (*Ficus carica* L.) leaves alleviated allergy and AD symptoms in cultured cells and animals. Therefore, here, we conducted a double-blind, randomized, controlled study in patients with mild AD to evaluate the safety and AD-relieving effects of prolonged consumption of fig leaf tea. Positive effects of fig leaf tea consumption were confirmed in 14 of 15 participants. Eczema Area and Severity Index values were significantly lowered in the fig leaf tea-treated group than in the placebo-treated group. The effect weakened 4 weeks after the end of the intervention, suggesting that continued intake of fig leaf tea was effective. Further assessments confirmed the safety of fig leaf tea consumption and revealed no variations that might pose a health hazard. Therefore, we postulate that fig leaf tea is a natural and safe therapeutic option for AD.

## 1. Introduction

Atopic dermatitis (AD) is a chronic, recurrent pruritic skin disease. The number of people with AD in Japan was approximately 510,000 in 2017, with a particularly high incidence in younger age groups. The causes of AD include abnormalities in the skin barrier, immune function, and lifestyle [1]. These various factors make it difficult to cure AD completely. Symptoms of AD are generally relieved by drug treatment, but the prolonged duration of treatment and the risk of side effects can be burdensome for patients. Therefore, there is a need for new drugs and alternatives that have fewer side effects even when taken over a long period of time. One of the safest AD therapies is the use of probiotics [2,3,4,5]. Although the efficacy of probiotics have been confirmed, the National Center for Complementary and Integrative Health and several reports have noted that probiotics or contaminating bacteria can cause other infections and their safety has not been well-established [6,7]. Therefore, it is very important to explore new AD treatment methods.

The main symptom of AD is itchy eczema. Epidermal keratinocytes, responsible for the barrier function of the skin, produce the filaggrin protein, and mutations in the filaggrin-encoding gene are a major risk factor for AD [8]. Abnormalities in filaggrin function allow foreign substances such as antigens and microorganisms to enter the body, causing allergic reactions in vivo. This allergic reaction is a type I allergy involving immunoglobulin (Ig)E antibodies and T helper (Th)2 lymphocytes. Chemicals such as histamine released from mast cells and Langerhans cells upon type I allergy cause itching and inflammation. AD patients with filaggrin gene mutations show high serum IgE levels and Th2 dominant immune responses [9]. Skin lesions in inflammation produce chemokines such as thymus and activation-regulated chemokine (TARC/CCL17) and macrophage-derived chemokine (MDC/CCL22), causing Th2 cells infiltrations into the lesion [10]. Infiltrating Th2 cells produce interleukin (IL)-4 and IL-13, which promote skin itching, thickening of skin, and decreased filaggrin production [11]. Furthermore, physical damage from skin scraping induces the expression of IL-1 and tumor necrosis factor-α, thus exacerbating inflammation [12,13]. Therefore, breaking the itch-scratch cycle is important in the treatment of AD. Medications for AD include topical steroids and tacrolimus or delgocitinib ointment. The efficacy and safety of these agents have been investigated in many clinical studies, and markedly strong therapeutic effects have been reported [14,15,16]. However, topical steroids reportedly cause side effects such as hypertension and hyperlipidemia and should be used with caution [17,18]. Tacrolimus and delgocitinib suppress the immune system and increase the risk of skin infections, which raises concerns regarding their use [19,20].

Fig (*Ficus carica* L.) belongs to the fruit tree family Moraceae, and its fruits are eaten fresh and also processed into other products. Consumption of fig fruit is reported to suppress DSS-induced colitis and relieve constipation [21,22]. Fig leaves have been used in traditional and Chinese medicine for a long time and exhibit various medical properties [23]. In vivo, fig leaf decoction is reported to lower blood glucose levels in both type I and type II diabetes and improve lipid profiles in high-fat-diet-fed rats [24,25,26,27]. In vitro, fig leaf extracts are reported to exhibit activity against liver, cervical, and breast cancers [28,29,30].

We previously investigated the functional properties of fig leaves and their processed tea. The leaves contain polyphenols such as caffeoylmalic acid and rutin, which have antioxidant properties [31]. Furthermore, in vivo and in vitro studies show that consumption of fig leaf tea alleviates type I allergy and AD [32]. In cells, fig leaf tea inhibits the binding of IgE antibodies to the FcεRI receptors. [32]. Omalizumab, a treatment for asthma, inhibits the binding of IgE antibodies to the FcεRI receptor. In recent years, the alleviation of AD following omalizumab treatment has been reported [33,34,35]. Based on these reports, fig leaf tea may therapeutic effects in AD. Although various benefits of fig leaves and tea have been reported, their consumption as food in Japan is limited to only a few regions. The consumption of fig leaf tea as a natural therapeutic agent against AD can be popularized to benefit public health. Therefore, in this study, we evaluated the safety and AD-relieving effects of prolonged fig leaf tea consumption in patients with mild AD.

## 2. Materials and Methods

### 2.1. Participants and Ethics Approval

This clinical study was conducted with volunteers registered at Miura Clinic, Medical Corporation Kanonkai (Osaka, Japan). The inclusion criteria for this study were as follows: (1) age between 20 and 59 years, (2) symptoms of AD in the range of 2–15 on the Eczema Area and Severity Index (EASI) evaluated visually (range 0–72) by a dermatologist, and (3) written informed consent given to participate in the study. The exclusion criteria were as follows: (1) history of serious illness such as diabetes, renal, hepatic, or gastrointestinal disease, (2) abnormal laboratory values for hepatic and renal functions, (3) history of gastrointestinal tract surgery, (4) currently undergoing any treatment, (5) use of oral medications or topical steroids for AD, (6) food and drug allergies, (7) anemia symptoms, (8) excessive exercise and weight loss, (9) consumption of dietary supplements, quasi-drugs, or medications for AD or allergic symptoms, (10) excessive alcohol consumption, (11) pregnancy or planned pregnancy during the study period, (12) participating in other clinical trials, or (13) other individuals deemed ineligible by the investigators.

The study was conducted in accordance with the ethical principles based on the Declaration of Helsinki (revised in 2013) and the main purpose of the “Ethical Guidelines for Medical Research Involving Human Subjects” (partially revised in 2017) after approval by the Ethical Review Committee of Miura Clinic, Medical Corporation Kanonkai (approval number: R1911 and approved 20 February 2020). The doctor explained the purpose, methods, and expected results of the study to the participants and obtained their written informed consent. The study protocol was registered with the University hospital Medical Information Network (UMIN; registration number: UMIN000039753).

### 2.2. Study Design

This was a double-blind, parallel two-arm study conducted from March to July 2020. The number of participants was calculated with an estimated effect size of 3.3, which corresponds to 60% of the EASI criterion of 5.5 at the time of selection, and a standard deviation of 3.5, because there was no precedent for a study of mild AD symptoms. The number of participants required was estimated to be 18 per group but was set at 15 per group considering the feasibility of the study. A staff member not directly involved in the study randomly assigned the participants to two study groups, namely, the fig leaf tea and placebo (colored water) intake groups, to prevent bias. The allocation method used was stratified randomization with an allocation ratio of 1:1. Assignment factors were gender, age, and EASI value. All but the person in charge of allocation were blinded to the allocation list until the data were collected and made public. However, in case of serious adverse events, necessary information was made available for disclosure. The study period was 12 weeks, with an 8-week intervention period and a 4-week non-intervention observation period. During the intervention period, the participants were asked to ingest 500 mL of either fig leaf tea or a placebo daily. The dose of fig leaf tea was set at 0.6° Bx, 500 mL/day, as this dose could be taken continuously without any discomfort. The intervention period was set at 8 weeks, the maximum period of time that could be supplied drinks at the set concentrations and volume. The participants visited the clinic before intervention (week 0), at weeks 4 and 8, and during the non-intervention observation period (week 12) and examined (Figure 1).

### 2.3. Preparation of Fig Leaf Tea

Fig leaf tea was prepared from the leaves of ‘Griśe de Tarascon’ (‘Dauphine’ *sensu* Condit) a fig cultivar that does not contain furanocoumarin [36,37]. Raw material was collected from our orchard (Kawanishi, Hyogo, Japan). Harvested leaves were washed, cut, steamed at 90 °C for 4 min, and dried at 60 °C for 4 h [32]. An infusion was prepared by macerating 12 kg of tea leaves in 80 °C water for 25 min. Fig leaf tea were adjusted to 0.6° Bx by adding water to the infusion (equivalent to 30 mg/mL). As a placebo, water with two natural dyes and two synthetic flavors was used. The natural dyes were 200 ppm Carsamus Yellow (Fujifilm Wako Pure Chemicals, Osaka, Japan), 7.5 ppm cochineal (Fujifilm Wako Pure Chemicals), and the synthetic flavors were 100 ppm “Black Tea Flavor” (T&M, Chiba, Japan) and 50 ppm “Oolong Tea Flavor” (T&M). Both samples were sterilized by ultra-high temperature sterilization after adding 500 ppm sodium ascorbate, 500 mL of each solution was aseptically filled into polyethylene terephthalate bottles. Sterilization conditions were set at 135 ± 1 °C at the holding tube outlet, with a sterilization time of 48 s, and an F0 value of 15.6 (Figure 2).

### 2.4. Study Outcomes

EASI values were studied as the primary outcomes. One dermatologist evaluated the symptoms of all the participants according to the method described in the Atopic Dermatitis Practice Guidelines 2018 [20]. The Patient-Oriented Eczema Measure (POEM) and Visual Analogue Scale (VAS) questionnaires, which assessed itching and interference with daily activities, were treated as secondary outcomes. The POEM questionnaire consists of seven questions about symptoms of itching, sleep disturbance, bleeding, exudate, cracking, desquamation, and dryness due to eczema [38]. The VAS questionnaire asked patients to estimate the maximum severity experienced in the past week for skin conditions (e.g., itching, redness, dryness) and interference with quality of life (e.g., sleep, concentration, fatigue) on a 10-cm long horizontal line. Scores ranged from 0.0 (no impact) to 10.0 (maximum impact). In the VAS, the skin condition was evaluated with six items and the degree of interference with daily life was evaluated with four items in total. Blood component analysis was also performed for TARC, Th1/Th2 ratio, and total IgE level. Blood samples were collected at the hospital visits at weeks 0, 4, 8, and 12.

### 2.5. Safety Assessments

Safety assessments included blood tests (WBC, RBC, hemoglobin, hematocrit, platelets, uric acid, urea nitrogen, aspartate aminotransferase (AST), alanine aminotransferase (ALT), alkaline phosphatase (ALP), lactate dehydrogenase (LDH), gamma-GTP, total bilirubin, total protein, albumin, creatinine (Cre), amylase, total cholesterol (Total-Cho), low-density lipoprotein cholesterol (LDL-Cho), triglycerides (TG), glucose, Na, K, Cl, Ca, Mg and Fe) and urine analysis (protein, sugar, urobilinogen, bilirubin, occult blood, ketone bodies, specific gravity, and pH). Blood and urine samples were collected at the time of selection (–2 weeks) and during the hospital visits at 8 and 12 weeks.

### 2.6. Nutrient Analysis of Fig Leaf Tea

The main components of fig leaf tea are shown in Table 1. Analysis of general nutritional composition (water, protein, fat, ash, carbohydrate, dietary fiber, and calories) was performed using methods based on the Standard Tables of Food Composition in Japan, 2020 edition [39]. The amount of total phenolic compounds was determined using the Folin–Ciocalteau method with catechin as a standard. Various flavonoids (rutin, quercetin 3′-(6′malonyl)-glucoside [Q3MG], isoshaftoside [ISS], and 8-hydroxycoumarin [8-HC]) were measured using liquid chromatograph-mass spectrometry/mass spectrometry (LC-MS/MS). For sample preparation, LC-20A (Shimadzu, Kyoto, Japan) with a Scherzo SM-C18 column (150 mm × 2 mm, 3 µm; Imtakt, Kyoto, Japan) was used and was maintained at 45 °C, with an injection volume of 5 µL and flow rate of 0.18 mL/min. To optimize the separation of the sample, two types of mobile phases were used including 0.3% formic acid-water (A) and 1.0% formic acid-acetonitrile (B). Gradient elution was performed as follows: 0–10 min, 0% B; 10–15 min, increase to 100% B; 15–25 min, maintained at 100% B; 25–35 min, maintained at 0% B. Quadruple time-of-flight MS (Q-TOF MS) (micrOTOF Q II; Bruker Daltonics, Billerica, MA, USA) was used for further analysis. Ionization conditions were as follows: electrospray ionization (ESI) method, measured mass range: 50–1500 *m*/*z*, capillary voltage: 4500 V for cations/2800 V for anions, nebulizer gas: N_2_ (1.6 bar), drying gas: N_2_ (7.0 L/min at 200 °C), quadrupole ion energy: 5.0 eV, collision energy: 10.0 eV. Mass spectra were analyzed through Compass Data Analysis (Bruker Daltonics). Flavonoid standard reagents were purchased as follows: rutin from Fujifilm Wako Pure Chemicals, Q3MG from Merck (Darmstadt, Germany), ISS from Funakoshi (Tokyo, Japan), 8-HC from MedChemExpress (Monmouth Junction, NJ, USA).

### 2.7. Statistical Analysis

Data are expressed as mean ± standard deviation. The *t*-test or Mann–Whitney U test was used to compare the fig leaf tea- and placebo-treated groups. Repeated measures ANOVA or Friedman test was used to compare the variation (Δ measurements) at each time point relative to the values before intervention. The significance level was set at *p* < 0.05.

All statistical analyses and sample size estimation were performed using EZR (ver. 2.7.–1), a modified version of R commander designed to add statistical functions frequently used in biostatistics [40].

## 3. Results

### 3.1. Background of Participants

Subject sampling and study profile for this study are summarized in Figure 3. Between 17 March 2020 and 23 March 2020, 30 individuals were selected from 88 candidates. The subjects were assigned to the fig leaf tea group and the placebo group randomly to equalize the EASI. General characteristics of the participants are listed in Table 2. Baseline characteristics such as age, gender, height, weight, body mass index, and pre-intervention EASI were not significantly different between the two groups. The EASI for both groups was set to be around 5.5, which is classified as mild in the severity assessment described by Leshem et al. [41]. The ingestion rate at the end of the study was 98.6% in both groups, and there were no cases of dropout or discontinuation during the study period. Therefore, all the participants were included in the analysis.

### 3.2. Comparison of Skin Symptoms

There was an improvement in the EASI in the fig leaf tea group compared with that in the placebo group (Table 3). The change in EASI in the placebo group was 0.2 ± 2.3 at week 4 and −0.2 ± 2.9 at week 8, whereas that in the fig leaf tea group was −1.7 ± 2.3 at week 4 and −3.1 ± 3.2 at week 8, indicating a significant decrease in the fig leaf tea group. No significant differences were observed between the two groups in week 12.

In the fig leaf tea group, EASI values were significantly lower at weeks 4 (*p* = 0.028) and 8 (*p* = 0.012) than before intervention (Figure 4A). Investigation of EASI transition in individual participants revealed that seven participants in the placebo-treated group exhibited worsening symptoms by week 8 compared with those before the intervention, whereas only one participant in the fig leaf tea-treated group showed worsening symptoms (Figure 4B); this participant showed no improvement during the intervention period (0–8 weeks).

### 3.3. Comparison of Subjective Symptoms

No significant differences were observed between the two groups using the POEM or VAS questionnaires, which are self-assessments of eczema in AD patients (Table 4 and Table 5). Before intervention, the POEM for the fig leaf tea group was 9.8 ± 5.9 and that for the placebo group was 13.1 ± 5.7, showing a trend toward lower values in the fig leaf tea group (*p* = 0.058). However, 8 weeks after intervention, there was no significant difference between the two groups (*p* = 0.900).

POEM in the placebo group decreased significantly at weeks 8 (*p* = 0.045) and 12 (*p* = 0.028) compared to that before intervention. However, VAS showed no variation in either skin symptoms or interference with QOL between the two groups during the intervention period.

### 3.4. Comparison of Blood Components

To compare the changes in blood components associated with immune responses, the blood TARC levels, Th1/Th2 ratio, and total IgE levels in the participants were measured (Table 6).

The TARC levels were markedly higher in one patient in the placebo group (week 0:10,990 pg/mL), but the other items did not show any significant changes, and thus data were included in the analysis. No significant differences were observed between the two groups in terms of TARC levels. In the placebo group, a significant increase in TARC levels was observed at week 4 compared to that before intervention (*p* = 0.009). However, no changes in TARC levels were observed in the fig leaf tea group before and during intervention (*p* = 0.058).

The Th1/Th2 ratio was higher in the fig leaf tea group than in the placebo group, but there were no significant differences in measured and variation values.

Total IgE levels increased during the study period in the placebo group, but decreased in the fig leaf tea group. However, the levels were not significantly different between the two groups.

### 3.5. Comparison of Safety Assessments

The fig leaf tea group showed a trend toward decreased blood levels of enzymes found in hepatocytes, such as AST, LDH, and ALP (Table 7). AST levels significantly decreased in the fig leaf tea group (−1.7 ± 3.3 U/L) compared with those in the placebo group (1.1 ± 3.2 U/L) at week 8 (*p* = 0.022). There was a significant decrease in LDH levels in the fig leaf tea group (−10.4 ± 16.6 U/L) compared to those in the placebo group (4.3 ± 19.9 U/L) at week 8 (*p* = 0.037). In addition, ALP levels were significantly lower in the fig leaf tea group at week 8 compared to thosebefore intervention (*p* = 0.015). For chloride (Cl), the measured value at week 8 was significantly higher in the fig leaf tea group (104.87 ± 1.13 mEq/L) than in the placebo group,(103.27 ± 1.79 mEq/L) (*p* = 0.007, Table 8).

## 4. Discussion

In this study, the effect of fig leaf tea consumption on AD was assessed through a human intervention study. EASI value is an objective measure recommended by the Harmonizing Outcome Measures for Eczema (HOME) to standardize clinical trial outcomes and is determined by a dermatologist’s visual assessment of AD symptoms in patients. We found that ingestion of fig leaf tea significantly reduced EASI values, indicating that fig leaf tea alleviates AD symptoms. As the effect of fig leaf tea was observed from the 4th week after intake, it was proven that the effect could be obtained if the intake was continued for at least one month. Moreover, EASI values increased 4 weeks after the end of fig leaf tea ingestion. Thus, the effects persisted for less than a month, and prolonged intake was recommended. Symptoms worsened during the intake period in seven participants in the placebo group, but only one participant in the fig leaf tea group. The study only required the consumption of 500 mL/day of the provided sample and did not restrict lifestyle practices (e.g., time of intake, caloric intake, moderate alcohol consumption, exercise, sleep duration, etc.). Nevertheless, the fact that EASI was reduced in most of the subjects in the fig leaf tea group indicated that the improvement was little influenced by diet and lifestyle.

Subjective symptoms were assessed using the POEM and VAS questionnaires. The POEM questionnaires focused on the evaluation of the patient’s own eczema, whereas the VAS questionnaires evaluated itching and decreased quality of life. In POEM responses, both groups reported a decrease in values and no significant difference was identified, but the placebo group reported a larger decrease (maximum decrease of 4.9). In POEM, a total variation of 3.4 or greater is considered clinically significant [42]. Thus, participants in the placebo group were clearly aware of the improvement in subjective symptoms. We speculate that the flavors added to the placebo samples had a relaxing effect. However, no significant improvement in skin condition was observed in the VAS for both groups. In the VAS, unlike the POEM results, the placebo group did not seem to perceive a marked improvement in skin condition. Since POEM and VAS are self-assessed, consistent evaluation is difficult, and therefore, differences were considered to have occurred between the two test methods. This difference could be avoided by studying a larger population.

Blood tests were conducted to compare blood levels of TARC, Th1/Th2 ratios, and total IgE, which are involved in immunity. TARC is a ligand for the chemokine receptor CCR4, which binds to CCR4-expressing Th2 cells, causing Th2 cells to migrate to the inflammatory site and worsen AD symptoms [10]. The amount of TARC in the blood of AD patients is associated with the severity of symptoms and is treated as a biomarker of AD symptoms [43,44,45]. In this study, no significant variation was observed between the two groups in terms of TARC levels. A possible reason could be the low EASI of the participants as a whole. In this study, it was necessary to recruit mild AD patients (EASI value of 15 or less) who could be classified as healthy individuals to comply with the Food with Functional Claims system in Japan. Because the EASI quantifies skin symptoms throughout the body, the values are lower if symptoms are localized. The low EASI values in both groups in the present study suggest that many of the participants had localized symptoms rather than symptoms throughout the body. In the present study, it was presumed that the improvement effect was not clear because the AD symptoms were localized and the initial TARC levels were low. One participant in the placebo group had notably high initial TARC levels, but the mean values for the other participants were below the normal range of 450 pg/mL (fig leaf tea group: 323 pg/mL; placebo group: 268 pg/mL). To confirm the effect of fig leaf tea on TARC levels, studies in patients with moderate AD with TARC levels of 750 pg/mL or higher or animal studies with controllable symptoms would prove useful.

The components of fig leaf tea that contribute to mitigating the effects of AD have not yet been identified. Fig leaf tea contains several flavonoids, such as rutin, Q3MG, ISS, and 8-HC. In AD mouse models, it is reported that rutin suppresses the induction of inflammatory cytokines such as IL-4 and IL-5, which promote allergic reactions, and thus reduce ear thickening and blood IgE levels [46]. In addition, a leaf extract of *Pyrus ussuriensis Maxim*, which mainly comprises rutin, has been reported to significantly suppress skin scratching and transepidermal water loss (TEWL) in mice [47]. A coumarin derivative, 7-hydoroxycoumarin (7-HC) reduces auricular thickening and inhibits inflammatory cytokines and chemokines in mice [48]. Fig leaf tea contains 8-HC, an isomer of 7-HC, which may have similar effects. Therefore, rutin and 8-HC probably contribute to the AD-relieving effects of fig leaf tea.

Th1/Th2 ratios were lower in the placebo group than in the fig leaf tea group before intervention. Because AD is an allergic disease that develops when the Th1/Th2 ratio decreases (Th2 cell type predominance), there was concern that the placebo group with a low ratio would be the group most likely to experience worsened symptoms. However, a comparison of the Th1/Th2 ratio in the two groups showed a decreasing trend in Th1 cell numbers in the placebo group, but no difference in Th2 cell numbers. Therefore, the decrease in Th1/Th2 ratio values in the placebo group was attributed to a decrease in Th1 cell numbers. Therefore, we determined that there was no difference in the predisposition to AD in the two groups. Total IgE levels increased in the placebo group, while they decreased in the fig leaf tea group throughout the study period. As total IgE levels increase or decrease depending on the predisposition of the individual to allergies and the external environment in addition to AD, it is difficult to discuss the results of this study in isolation; however, the inhibitory effect of fig leaf tea on the increase in IgE antibody levels may have likely contributed to the AD alleviation effect.

Laboratory tests performed as part of the safety assessments showed significant reductions in AST, LDH, and ALP levels upon fig leaf tea consumption. These enzymes are considered indicators of liver function, because they are abundantly expressed in hepatocytes and are released into the blood when hepatocytes are damaged. Although fig leaf extract has been reported to reduce blood levels of these enzymes in animal studies [25,49,50,51], this study is the first to show that the same effect is observed in humans. Although not much is known about the relationship between liver function and AD symptoms, it has been reported that in pediatric AD, a significantly higher percentage of patients have fatty liver and increased AST levels compared to healthy controls [52,53,54]. Liver disease is also known to cause intense generalized itching. Patients with liver disease exhibit increased plasma concentrations of histamine and µ-opioids, which are thought to contribute to hepatic pruritus [55,56,57,58]. Because AD is a type of allergic pruritus, antihistamines are widely used for its treatment. Furthermore, opioids are expressed in epidermal cells of AD patients, and Naloxone topical, a µ-opioid antagonist, has been reported to suppress itching [59,60]. Improvement of liver function upon treatment with fig leaf tea may have led to a reduction in blood levels of histamine and µ-opioids, which in turn may have led to the alleviation of AD symptoms.

We also observed increased blood Cl in the fig leaf tea group. Because increased Cl is associated with worsened renal function; there was a concern about renal function impairment due to fig leaf tea ingestion. However, no significant differences in blood creatinine (Cre) levels, a highly reliable indicator of renal function, were observed during the study period in both groups (Table 8). Therefore, it was concluded that prolonged intake of fig leaf tea had no effect on renal function. There were no significant differences in the levels of other blood components in the two groups, suggesting that fig leaf tea does not cause serious side effects. These results suggest that prolonged consumption of fig leaf tea without furanocoumarin does not cause serious adverse effects in the range of commonly measured blood constituents. However, many fig varieties are known to contain furanocoumarins, such as psoralen and bergapten. Psoralen is known to be phototoxic, increasing the effects of UV light, and cases of erythematous and edematous rashes have been reported upon treatment with fig leaves [61]. It has also been reported that furanocoumarin in grapefruit inhibits the activity of the drug-metabolizing enzyme CYP3A4, maintaining high drug concentrations in the blood [62,63,64]. The potential of furanocoumarin in figs to interact with drugs is high and caution should be exercised when drinking fig leaf tea containing furanocoumarin. For safe AD amelioration, it is recommended to consume tea made from fig varieties that do not contain furanocoumarin or those from which furanocoumarin has been removed.

This study is valuable in that it is was a randomized, double-blind, subject-controlled trial that established that fig leaf tea consumption is beneficial in improving the dermatitis symptoms of AD. This study is limited in that the sample size was small. The sample size estimate based on the value of change at week 8, when a significant difference was confirmed (difference in means between the two groups; 2.94, overall standard deviation; 3.35), showed that 21 participants per group was an appropriate sample size. This estimate focused only on the EASI value. In complex diseases such as AD, it is necessary to evaluate not only EASI values but also TARC values, TEWL, subjective symptoms, and various other disease signs to determine efficacy. Therefore, more trials with a larger number of subjects are needed to obtain statistically and clinically meaningful results on the effects of fig leaf tea in the future.

## 5. Conclusions

We evaluated the safety and AD-relieving effects of prolonged fig leaf tea consumption. Prolonged consumption of fig leaf tea was shown to significantly reduce EASI values. Fig leaf tea use was largely found to be safe. Therefore, prolonged consumption of fig leaf tea may be a safe and effective alternative to current therapies for AD. The findings of this study must be further validated in a larger sample size.

## Figures and Tables

**Figure 1 nutrients-14-04470-f001:**
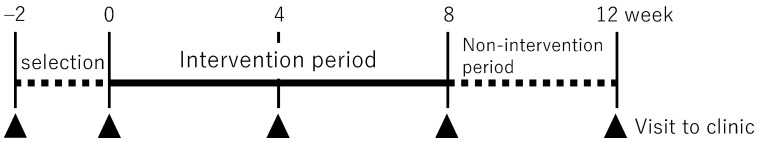
Intervention test schedule. An 8–week intake period and a 4–week non-intake period were established. Various tests were performed at 0, 4, 8, and 12 weeks.

**Figure 2 nutrients-14-04470-f002:**
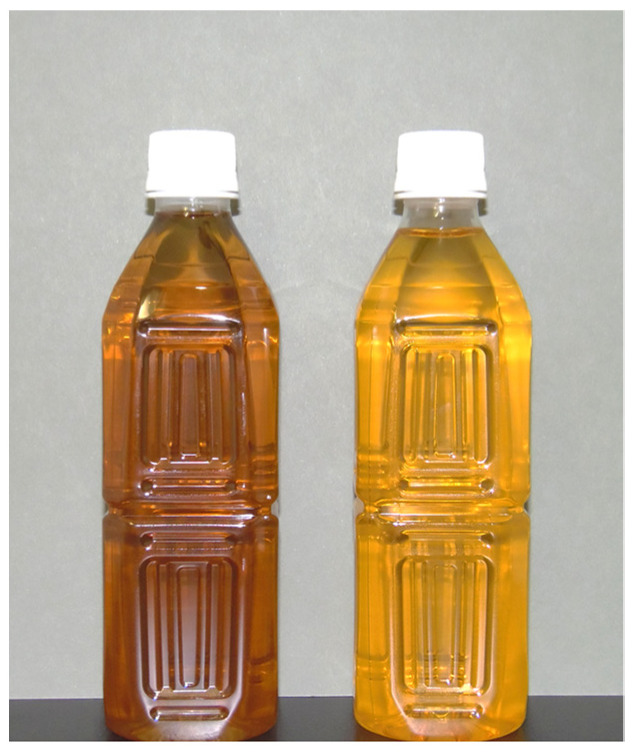
Appearance of test beverages. Fig leaf tea (left) and placebo (colored water: (right)) served in 500 mL PET bottles.

**Figure 3 nutrients-14-04470-f003:**
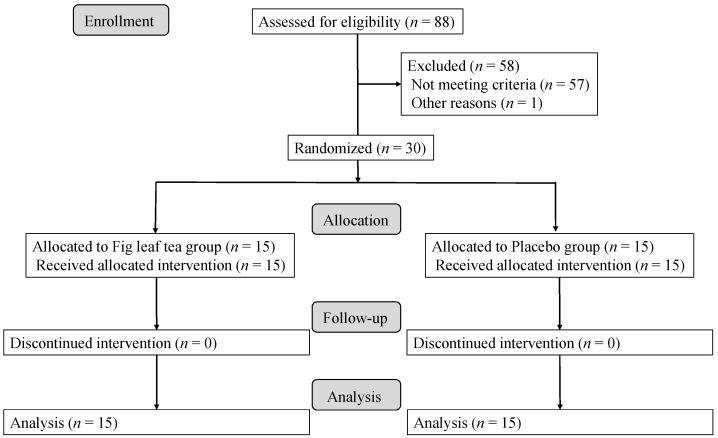
Flow chart for the study subjects. The number of participants enrolled, allocated, followed and analyzed, is shown using CONSORT-SPI 2018 Flow Diagram.

**Figure 4 nutrients-14-04470-f004:**
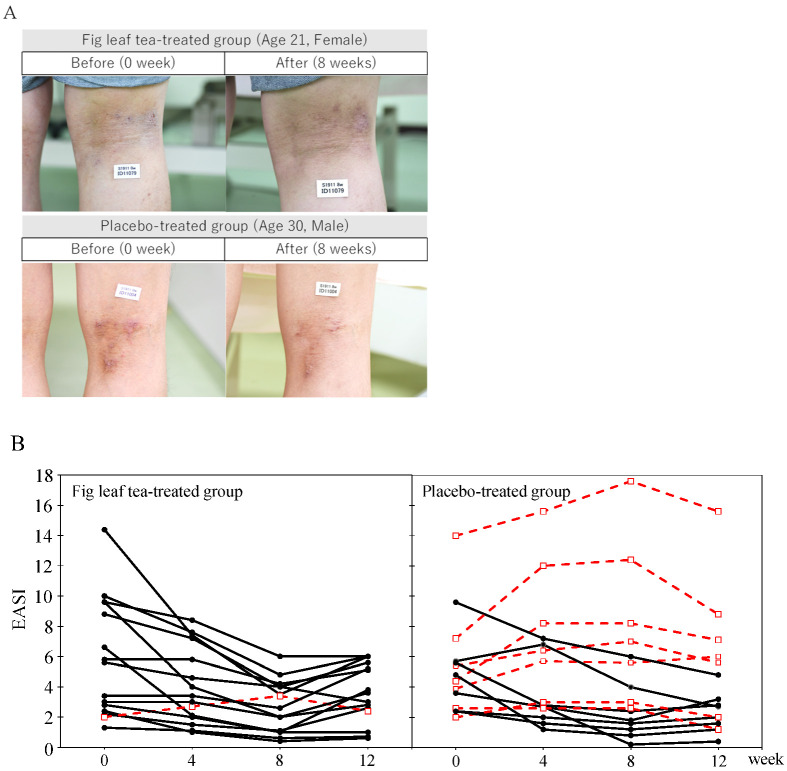
Changes in AD Symptoms and Individual EASI transitions. (**A**): Representative skin condition at 0 and 8 week in subjects from fig leaf tea and placebo group. (**B**): Changes in EASI per subject during the study period. Red dashed lines indicate subjects whose EASI at the end of intervention (8 weeks) was worse than before intervention (0 weeks).

**Table 1 nutrients-14-04470-t001:** Nutritional composition of fig leaf tea.

Nutritional Compositions	Contents
Calories (Kcal/100 mL)	2
Carbohydrate (g/100 mL)	0.5
Crude fat (g/100 mL)	<0.1
Crude protein (g/100 mL)	<0.1
Water (%)	99.8
Ash (g/100 mL)	<0.1
Fiber (g/100 mL)	<0.1
Total phenolic compounds (g/100 mL)	0.02
Rutin (mg/100 mL)	3.7
Quercetin malonyl glucoside (mg/100 mL)	0.6
Isoschaftoside (mg/100 mL)	2.3
8-Hydoroxy coumalin (mg/100 mL)	1
Sodium ascorbate (mg/100 mL)	250

**Table 2 nutrients-14-04470-t002:** Background of study participants.

	Fig Leaf Tea Group (*n* = 15)	Placebo Group (*n* = 15)	*p*-Value
Age (years)	41.0 ± 10.8	40.7 ± 11.2	0.934
Sex (M/F)	8/7	9/6	1.000
Height (cm)	167.2 ± 7.7	166.3 ± 11.6	0.815
Weight (kg)	64.3 ± 11.7	65.4 ± 13.9	0.824
BMI (kg/m^2^)	23.0 ± 3.5	23.4 ± 3.0	0.696
EASI	5.83 ± 3.89	5.15 ± 3.18	0.803

M: male; F: female; BMI: body mass index. Values are presented as mean ± SD.

**Table 3 nutrients-14-04470-t003:** Transition of EASI values.

	Group	Measurement				Variation		
		0 Week	4 Week	8 Week	12 Week	4 Week	8 Week	12 Week
EASI	Fig leaf tea	5.83 ± 3.89	4.12 ± 2.56 ^†^	2.71 ± 1.71 ^†^	3.63 ± 1.96	−1.71 ± 2.25	−3.13 ± 3.18	−2.21 ± 2.92
	Placebo	5.15 ± 3.18	5.37 ± 4.14	4.96 ± 4.81	4.33 ± 3.95	0.22 ± 2.31	−0.19 ± 2.92	−0.81 ± 2.28
*p*-value		0.803	0.633	0.280	0.983	0.044 *	0.036 *	0.245

Values are presented as mean ± SD. * *p* < 0.05; Mann–Whitney U test to compare the placebo group. ^†^
*p* < 0.05; Friedman test to compare the 0 week value of each group.

**Table 4 nutrients-14-04470-t004:** Transition of POEM.

	Group	Measurement				Variation		
		0 Week	4 Week	8 Week	12 Week	4 Week	8 Week	12 Week
POEM	Fig leaf tea	9.8 ± 5.9	8.5 ± 4.9	8.5 ± 6.7	7.5 ± 7.1	−1.3 ± 5.8	−1.3 ± 7.4	−2.3 ± 7.1
	Placebo	13.1 ± 5.7	10.7 ± 4.7	8.3 ± 5.8 ^†^	8.3 ± 4.0 ^†^	−2.4 ± 3.9	−4.8 ± 5.2	−4.9 ± 4.9
*p*-value		0.058	0.220	0.900	0.453	0.544	0.109	0.176

Values are presented as mean ± SD. ^†^
*p* < 0.05; Friedman test to compare the 0 week value of each group.

**Table 5 nutrients-14-04470-t005:** Transition of VAS questionnaires.

VAS	Group	Measurement				Variation		
		0 Week	4 Week	8 Week	12 Week	4 Week	8 Week	12 Week
Skin condition	Fig leaf tea	29.1 ± 13.4	25.0 ± 13.5	22.3 ± 13.1	22.1 ± 16.2	−4.1 ± 10.5	−6.7 ± 12.2	−6.9 ± 14.8
	Placebo	32.1 ± 14.2	28.1 ± 14.1	23.9 ± 13.9	24.3 ± 14.3	−4.0 ± 11.3	−8.2 ± 11.6	−7.8 ± 11.3
*p*-value		0.663	0.520	0.917	0.683	0.724	0.806	0.806
Impact on QOL	Fig leaf tea	15.6 ± 8.2	16.2 ± 11.7	15.7 ± 11.4	16.9 ± 12.4	0.6 ± 7.8	0.1 ± 8.6	1.4 ± 8.2
	Placebo	18.2 ± 10.5	17.8 ± 10.0	17.0 ± 10.8	19.3 ± 11.7	−0.4 ± 9.4	−1.1 ± 8.8	1.2 ± 8.4
*p*-value		0.481	0.653	0.618	0.595	0.838	0.820	0.868

Values are presented as mean ± SD. Skin conditions are itching, redness, dryness etc. and Impact on QOL are interference with quality of life (e.g., sleep, concentration, fatigue etc.).

**Table 6 nutrients-14-04470-t006:** Transition in blood components levels.

	Group	Measurement				Variation		
		0 Week	4 Week	8 Week	12 Week	4 Week	8 Week	12 Week
TARC	Fig leaf tea	323 ± 146	399 ± 202	365 ± 169	406 ± 255	76 ± 94	42 ± 79	82 ± 175
	Placebo	983 ± 2770	1191 ± 2856 ^††^	808 ± 1620	671 ± 1240	208 ± 360	−175 ± 1217	−312 ± 1541
*p*-value		0.624	0.775	0.983	0.967	0.534	0.772	0.494
Th1/Th2	Fig leaf tea	8.2 ± 3.8	7.7 ± 3.6	7.5 ± 3.6	8.3 ± 3.4	−0.4 ± 3.3	−0.7 ± 1.6	0.2 ± 1.6
	Placebo	6.4 ± 2.6	6.4 ± 3.5	5.6 ± 2.4	6.4 ± 2.6	0.0 ± 2.4	−0.7 ± 1.6	0.0 ± 1.6
*p*-value		0.142	0.308	0.109	0.090	0.667	0.919	0.777
IgE	Fig leaf tea	314 ± 788	282 ± 647	294 ± 670	303 ± 706	−32 ± 145	−20 ± 124	−11 ± 96
	Placebo	297 ± 438	337 ± 606	406 ± 716	456 ± 924	41 ± 179	109 ± 287	160 ± 497
*p*-value		0.267	0.461	0.187	0.217	0.507	0.089	0.217

Values are presented as mean ± SD. ^††^
*p* < 0.01; Friedman test to compare the 0 week value of each group.

**Table 7 nutrients-14-04470-t007:** Transition in liver enzyme levels.

	Normal Range	Group	Measurement			Variation	
			−2 Week	8 Week	12 Week	8 Week	12 Week
AST	10–40	Fig leaf tea	19.2 ± 4.7	17.5 ± 2.5	17.8 ± 3.5	−1.7 ± 3.3	−1.4 ± 2.2
		Placebo	19.7 ± 6.4	20.8 ± 6.6	20.2 ± 7.1	1.1 ± 3.2	0.5 ± 3.1
*p*-value			0.820	0.078	0.250	0.022 *	0.061
ALT	5–40	Fig leaf tea	17.5 ± 10.1	15.5 ± 6.8	16.1 ± 9.7	−2.0 ± 5.6	−1.5 ± 4.1
		Placebo	19.6 ± 13.1	20.7 ± 10.5	19.9 ± 11.6	1.1 ± 6.6	0.3 ± 5.0
*p*-value			0.631	0.125	0.330	0.181	0.291
LDH	115–245	Fig leaf tea	169.7 ± 21.6	159.3 ± 19.2	166.4 ± 22.6	−10.4 ± 16.6	−3.3 ± 16.1
		Placebo	168.9 ± 40.4	173.3 ± 46.8	169.0 ± 41.2	4.3 ± 19.9	0.1 ± 11.9
*p*-value			0.951	0.293	0.832	0.037 *	0.525
ALP	115–359	Fig leaf tea	166.2 ± 35.4	157.7 ± 35.4 ^†^	156.9 ± 31.3	−8.5 ± 9.8	−9.3 ± 15.1
		Placebo	178.2 ± 43.8	180.0 ± 45.7	182.4 ± 39.6	1.8 ± 21.3	4.2 ± 24.7
*p*-value			0.417	0.148	0.060	0.105	0.083

Values are presented as mean ± SD. * *p* < 0.05; *t*-test to compare the placebo group. ^†^
*p* < 0.05; Repeated measure ANOVA to compare the −2 week value of each group.

**Table 8 nutrients-14-04470-t008:** Transition in safety assessments.

	Normal Range	Group	Measurement			Variation	
			−2 Week	8 Week	12 Week	8 Week	12 Week
uric acid	M: 3.7–7.0 F: 2.5–7.0	Fig leaf tea	5.03 ± 1.38	5.15 ± 1.37	5.15 ± 1.21	0.12 ± 0.57	0.12 ± 0.67
		Placebo	5.25 ± 1.47	5.60 ± 1.48	5.72 ± 1.68	0.35 ± 0.70	0.47 ± 0.74
*p*-value			0.667	0.391	0.293	0.338	0.188
urea nitrogen	8.0–22.0	Fig leaf tea	13.76 ± 3.84	14.26 ± 2.99	13.71 ± 2.07	0.50 ± 2.30	−0.05 ± 3.73
		Placebo	12.64 ± 2.95	12.11 ± 3.73	12.70 ± 3.41	−0.53 ± 3.74	0.06 ± 2.85
*p*-value			0.379	0.094	0.336	0.375	0.931
γ-GTP	M: ≤70 F: ≤30	Fig leaf tea	21.67 ± 12.53	24.40 ± 17.89	22.33 ± 14.82	2.73 ± 14.58	0.67 ± 3.87
		Placebo	21.33 ± 17.17	21.67 ± 16.24	21.93 ± 18.42	0.33 ± 4.10	0.60 ± 3.50
*p*-value			0.952	0.665	0.948	0.548	0.961
total bilirubin	0.3–1.2	Fig leaf tea	0.69 ± 0.27	0.73 ± 0.35	0.77 ± 0.31	0.03 ± 0.33	0.08 ± 0.44
		Placebo	0.63 ± 0.30	0.73 ± 0.33	0.66 ± 0.28	0.10 ± 0.23	0.03 ± 0.23
*p*-value			0.527	1.000	0.299	0.531	0.717
total protein	6.7–8.3	Fig leaf tea	7.39 ± 0.26	7.27 ± 0.34	7.29 ± 0.30	−0.12 ± 0.33	−0.11 ± 0.23
		Placebo	7.37 ± 0.29	7.43 ± 0.43	7.36 ± 0.34	0.06 ± 0.33	−0.01 ± 0.24
*p*-value			0.844	0.267	0.537	0.151	0.280
albumin	3.8–5.2	Fig leaf tea	4.54 ± 0.26	4.54 ± 0.32	4.56 ± 0.28	0.00 ± 0.23	0.02 ± 0.22
		Placebo	4.63 ± 0.22	4.69 ± 0.29	4.65 ± 0.20	0.05 ± 0.27	0.01 ± 0.20
*p*-value			0.324	0.199	0.344	0.565	0.932
creatinine	M: 0.61–1.04 F: 0.47–0.79	Fig leaf tea	0.73 ± 0.14	0.76 ± 0.14	0.75 ± 0.13	0.03 ± 0.08	0.02 ± 0.09
		Placebo	0.71 ± 0.18	0.72 ± 0.17	0.71 ± 0.16	0.01 ± 0.05	0.00 ± 0.05
*p*-value			0.685	0.453	0.465	0.453	0.549
amylase	37–125	Fig leaf tea	82.13 ± 15.56	79.47 ± 18.49	78.80 ± 18.00	−2.67 ± 11.63	−3.53 ± 8.80
		Placebo	76.33 ± 25.92	68.73 ± 23.86	70.73 ± 26.68	−7.60 ± 6.83	−5.60 ± 10.52
*p*-value			0.465	0.180	0.353	0.170	0.567
total-Cho	150–219	Fig leaf tea	197.13 ± 29.81	193.53 ± 31.71	191.27 ± 36.58	−3.60 ± 14.82	−5.87 ± 14.34
		Placebo	195.67 ± 35.35	193.80 ± 38.63	190.93 ± 36.72	−1.87 ± 15.56	−4.73 ± 13.91
*p*-value			0.903	0.984	0.980	0.757	0.828
LDL-Cho	70–139	Fig leaf tea	115.67 ± 21.97	115.20 ± 25.01	111.07 ± 27.54	−0.47 ± 19.96	−4.60 ± 15.57
		Placebo	117.40 ± 33.36	116.47 ± 32.44	115.00 ± 30.10	−0.93 ± 14.38	−2.40 ± 13.78
*p*-value			0.868	0.906	0.712	0.936	0.685
TG	50–149	Fig leaf tea	97.33 ± 53.10	77.33 ± 38.51	91.47 ± 67.02	−20.00 ± 27.72	−5.87 ± 56.67
		Placebo	102.07 ± 71.83	98.33 ± 61.22	90.33 ± 58.73	−3.73 ± 42.02	−11.73 ± 39.44
*p*-value			0.839	0.272	0.961	0.223	0.745
glucose	70–109	Fig leaf tea	86.33 ± 9.22	87.80 ± 9.45	89.00 ± 7.50	1.47 ± 4.91	2.67 ± 8.05
		Placebo	90.53 ± 10.00	88.73 ± 7.37	94.73 ± 12.54	−1.80 ± 6.05	4.20 ± 9.01
*p*-value			0.242	0.765	0.142	0.116	0.627
Na	136–147	Fig leaf tea	140.80 ± 1.01	140.07 ± 1.33	140.27 ± 1.87	−0.73 ± 1.16	−0.53 ± 1.46
		Placebo	140.27 ± 1.10	139.20 ± 1.93	139.47 ± 2.03	−1.07 ± 1.22	−0.80 ± 2.04
*p*-value			0.178	0.166	0.271	0.451	0.684
Cl	98–109	Fig leaf tea	103.93 ± 1.53	104.87 ± 1.13	104.33 ± 1.23	0.93 ± 1.28	0.40 ± 1.45
		Placebo	103.60 ± 1.72	103.27 ± 1.79	104.07 ± 2.55	−0.33 ± 2.26	0.47 ± 3.09
*p*-value			0.580	0.007 **	0.719	0.072	0.940
K	3.6–5.0	Fig leaf tea	4.05 ± 0.22	3.98 ± 0.13	3.94 ± 0.16	−0.07 ± 0.18	−0.11 ± 0.23
		Placebo	4.15 ± 0.18	3.91 ± 0.23	4.02 ± 0.30	−0.24 ± 0.28	−0.13 ± 0.31
*p*-value			0.158	0.340	0.370	0.055	0.789
Mg	1.8–2.6	Fig leaf tea	2.29 ± 0.12	2.29 ± 0.14	2.27 ± 0.13	−0.01 ± 0.21	−0.03 ± 0.14
		Placebo	2.25 ± 0.17	2.22 ± 0.18	2.17 ± 0.17	−0.03 ± 0.09	−0.03 ± 0.20
*p*-value			0.471	0.265	0.412	0.650	0.915
Ca	8.5–10.2	Fig leaf tea	9.45 ± 0.23	9.37 ± 0.35	9.36 ± 0.33	−0.07 ± 0.30	−0.09 ± 0.31
		Placebo	9.46 ± 0.23	9.55 ± 0.33	9.35 ± 0.24	0.09 ± 0.32	−0.11 ± 0.25
*p*-value			0.876	0.176	0.949	0.170	0.849
Fe	F: 54–200 M: 48–154	Fig leaf tea	108.87 ± 45.17	94.73 ± 52.37	108.73 ± 50.86	−14.13 ± 57.02	−0.13 ± 70.60
		Placebo	99.73 ± 33.67	99.93 ± 40.33	93.67 ± 42.32	0.20 ± 47.90	−6.07 ± 46.56
*p*-value			0.536	0.763	0.386	0.462	0.788

Values are presented as mean ± SD. ** *p* < 0.01; *t*-test to compare the placebo group.

## Data Availability

Not applicable.

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
