# Peer review of "Efficacy and Safety of Fig (Ficus carica L.) Leaf Tea in Adults with Mild Atopic Dermatitis: A Double-Blind, Randomized, Placebo-Controlled Preliminary Trial"

_nutrients, 2022, doi:10.3390/nu14214470_

Round 1

Reviewer 1 Report

The manuscript investigated the potential of fig tea consumption to relieve the symptoms of mild AD in a double-blind, randomized, placebo-controlled trial. The safety of fig tea consumption was also evaluated. The results were shown that the EASI values were significantly reduced after fig tea intervention, and it was not cause harmful effects on the body within the tested range.However, the manuscript needs to be modified as follows:

1. The language logic was confused in the introduction (lines 39-57), please rewrite it.

2. Please supplement the connection between fig and AD in introduction.

3. Please describe in detail how the dose and intervention time of fig tea consumption were obtained.

4. The degree of intervention of fig tea was different in every participants (Figure 4B). Meanwhile, only 30 people took part in this trail. The result doesn't seem to support your opinion in discussion (lines 305-307), which describedThese results suggest that the effects of fig tea vary little among individuals, and it is expected to act without being affected by dietary or lifestyle habits.Please explain it.

Author Response

18-Oct-2022

Dear Reviewer 1

I wish to express my appreciation to the Reviewer for your insightful comments, which have helped me significantly improve the paper.

The manuscript investigated the potential of fig tea consumption to relieve the symptoms of mild AD in a double-blind, randomized, placebo-controlled trial. The safety of fig tea consumption was also evaluated. The results were shown that the EASI values were significantly reduced after fig tea intervention, and it was not cause harmful effects on the body within the tested range.However, the manuscript needs to be modified as follows:

Response: We thank the Reviewer for this pertinent comment. We have addressed the points you raised as follows. We hope these answers meet your expectations.

  1. The language logic was confused in the introduction (lines 41-62), please rewrite it.

Response: As the reviewer said, the part pointed out was logically confusing. Therefore, we have rewritten this part of the report to focus on the mechanism of eczema, a symptom of AD, and its treatment.

  1. Please supplement the connection between fig and AD in introduction.

Response: Fig tea and omalizumab have similar mechanisms of allergy suppression, and the AD-improving effect of omalizumab led us to the AD-relieving effect of fig tea. Please check the additional description (lines 90–94).

  1. Please describe in detail how the dose and intervention time of fig tea consumption were obtained.

Response: We set the doses based on the subject's ability to continue taking the product. Please check the additional description for details (lines 138–141).

  1. The degree of intervention of fig tea was different in every participants (Figure 4B). Meanwhile, only 30 people took part in this trail. The result doesn't seem to support your opinion in discussion (lines 305-307), which described“These results suggest that the effects of fig tea vary little among individuals, and it is expected to act without being affected by dietary or lifestyle habits”.Please explain it.

Response: We agree your comment. The sample size is too small to consider small individual differences. However, since fig leaf tea reduced EASI in the subjects' own lifestyle, we believe that fig leaf tea acted independently of lifestyle. Therefore, the notation "little among individual" was removed, but "independent of lifestyle" was retained (lines 340-345).

Sincerely,

Tatsuya Abe

Reviewer 2 Report

In their manuscript, Abe et al. describe a placebo controlled trial to test the ability of tea made from fig leaves to ameliorate atopic dermatitis. The test is claimed to be double blinded and comprised 15 individuals in both groups. The administration of tea or placebo was for 2 weeks and testing was performed during application and one week afterwards. The measurements are carefully performed and the manuscript is well written. Unfortunately, heterogeneous results were obtained and only weak positive effects could be observed if at all in the fig leave tea group. Nevertheless, I feel the manuscript should be considered as a first approach and stronger positive effects might be observed when the study is extended.

1.      M&M line 122 123: explain what adjusted to 0.60Bx indicates.

2.      The placebo group received a mixture of aqueous salt solution flavored with extracts of two teas compare with the fig leave group. Since the probands can taste what they receive one cannot real consider the administration as blind. This should be discussed.

3.      The random selection does not look very random. This is most likely due to the low numbers of test persons. This should also be discussed.

4.      It appears to me naiv to expect a strong effect when applying an aqueous extract orally against a complex disease like atopic dermatitis. Most likely the active compound is found only in low concentration and requires to be efficiently taken up and find the right anatomical location. Therefore, the study needs to be seriously extended over time. This is mentioned in the discussion but should be pointed out more strongly.

5.      I also challenge the point that an increase to 21 probands would be sufficient to obtain meaningful results. The authors deal with a highly complex subject. To receive statistically and clinically meaningful results the number of test persons needs to be very much increased. This should also be discussed.

Author Response

18-Oct-2022

Dear Reviewer 2

I wish to express my appreciation to the Reviewer for your insightful comments, which have helped me significantly improve the paper.

In their manuscript, Abe et al. describe a placebo controlled trial to test the ability of tea made from fig leaves to ameliorate atopic dermatitis. The test is claimed to be double blinded and comprised 15 individuals in both groups. The administration of tea or placebo was for 2 weeks and testing was performed during application and one week afterwards. The measurements are carefully performed and the manuscript is well written. Unfortunately, heterogeneous results were obtained and only weak positive effects could be observed if at all in the fig leave tea group. Nevertheless, I feel the manuscript should be considered as a first approach and stronger positive effects might be observed when the study is extended.

Response: We thank the Reviewer for this pertinent comment. As the reviewer said, unfortunately, this trial did not show strong or consistent results. We believe that the strength of food is that it can be taken safely and continuously over a long period of time, even if it does not have a strong effect like a drug in a short period of time. Although this study was limited in terms of sample size and duration, we expect that the effectiveness of fig leaf tea will become clearer as these issues are resolved in the future.

  1. M&M line 122 123: explain what adjusted to 0.60Bx indicates.

Response: We rewrote "The infusion was adjusted to 0.6ºBx by adding water" to "Fig leaf tea were adjusted to 0.6ºBx by adding water to the infusion". Please check the additional description (lines 152–153).

  1. The placebo group received a mixture of aqueous salt solution flavored with extracts of two teas compare with the fig leave group. Since the probands can taste what they receive one cannot real consider the administration as blind. This should be discussed.

Response: We apologize for the misleading way in which we have written this. All flavors used in this project are chemically synthesized. Those product names were "Black Tea Flavor" & "Oolong Tea Flavor". Therefore, the placebo sample does not contain tea-derived components (line 154–157).

  1. The random selection does not look very random. This is most likely due to the low numbers of test persons. This should also be discussed.

Response: For this study, participants were required to be able to visit the hospital every two weeks and to receive regular mailings of samples. We considered people who fit these requirements to be those who live in the vicinity of the implementation facility. Therefore, we recruited participants who met the other test criteria in the suburbs, which resulted in a total of 88 participants. We do not consider 88 participants in the selection to be particularly small for a 30-person selection. In future examinations, we would like to set the number of people to be selected in consideration of the points you have raised.

  1. It appears to me naiv to expect a strong effect when applying an aqueous extract orally against a complex disease like atopic dermatitis. Most likely the active compound is found only in low concentration and requires to be efficiently taken up and find the right anatomical location. Therefore, the study needs to be seriously extended over time. This is mentioned in the discussion but should be pointed out more strongly.
  2. I also challenge the point that an increase to 21 probands would be sufficient to obtain meaningful results. The authors deal with a highly complex subject. To receive statistically and clinically meaningful results the number of test persons needs to be very much increased. This should also be discussed.

Response to comments 4 and 5: We think both comments are very important points. As the reviewer pointed out, we do not think that the present data are sufficient to determine the effect of fig leaf tea on AD mitigation. On the other hand, the results of this study are the first to show that fig leaf tea contributes to AD mitigation in humans. Therefore, we consider and hope that these results are significant enough to publish in a paper. In the discussion, we wrote that 21 was the optimal number of participants, but this statement is based solely on the focus on the EASI variation of the study. Since AD is a complex disease, we need more trials and discussion on the effectiveness of fig leaf tea. Please confirm the above information in the text (lines 449–454).

Sincerely,

Tatsuya Abe
